# Actuators for Implantable Devices: A Broad View

**DOI:** 10.3390/mi13101756

**Published:** 2022-10-17

**Authors:** Bingxi Yan

**Affiliations:** Department of Electrical and Computer Engineering, Ohio State University, Columbus, OH 43210, USA; yan.575@buckeyemail.osu.edu

**Keywords:** biomedical, implantable robots, drug-delivery capsule, micro-swimmer, stent, catheter

## Abstract

The choice of actuators dictates how an implantable biomedical device moves. Specifically, the concept of implantable robots consists of the three pillars: actuators, sensors, and powering. Robotic devices that require active motion are driven by a biocompatible actuator. Depending on the actuating mechanism, different types of actuators vary remarkably in strain/stress output, frequency, power consumption, and durability. Most reviews to date focus on specific type of actuating mechanism (electric, photonic, electrothermal, etc.) for biomedical applications. With a rapidly expanding library of novel actuators, however, the granular boundaries between subcategories turns the selection of actuators a laborious task, which can be particularly time-consuming to those unfamiliar with actuation. To offer a broad view, this study (1) showcases the recent advances in various types of actuating technologies that can be potentially implemented in vivo, (2) outlines technical advantages and the limitations of each type, and (3) provides use-specific suggestions on actuator choice for applications such as drug delivery, cardiovascular, and endoscopy implants.

## 1. Introduction

Actuators generate motion. Skeletal muscle, as the best known bio-actuators driving bio-movement in nature, consists of bundled fascicles that can further break down into highly contractile units called myofibrils [1]. Blood vessels and motor neurons are integrated inside to regulate metabolite circulation and bioelectrical transport [2]. This muscle–vessel–neuron trinity includes three core elements of actuation, power, and control, a principle human-made actuators follow. Among the three, the choice of actuator seems less challenging comparing to the other two, given the apparent value of high-performance tactile sensors for dexterous robotics [3,4], and the value of lightweight powering for aerosol insect-mimic flying robotics [5,6,7]. However, most actuators reported to data are designed for ex vivo applications and may fail to undertake tasks in vivo due to mechanical, biochemical, or thermal mismatch with the surrounding bio-ambient conditions.

The past decade has seen a rapidly expanding library of biomedical actuators in piezoelectric [8,9], electro-/thermo-active [10,11], magnetic, and pneumatic materials [12,13]. Each category provides technical advantages for certain applications. For example, pneumatic actuators can generate a large strain above 300% but only work at a low frequency [14]. By contrast, electro-actuators can provide frequency over 1 kHz for high-speed actuation [15]. In addition to the technical specifications of the actuator itself, the selection of an actuator involves considerations of power and control. For instance, implementing pneumatic and hydraulic actuations in the body requires a water or air supply via a catheter or cable, yet magnetic or electric actuators can be powered in a completely cable-free approach. Moreover, choosing an actuator for in vivo applications often involves the consideration of additional aspects including durability and biocompatibility with the surrounding tissues [16,17,18]. Most of the existing reviews focus on one subcategory such as pneumatic, electroactive, or optical actuators [19,20,21,22,23], and their applications in drug delivery, thrombus removal, or ventricular assist devices (VADs). Readers often need to go through each subcategory to acquire a full vision for selecting an actuator. For example, the blood-circulating pump in VADs usually consist of battery-powered motors and rotors. Seeing a good match with respect to the stress and frequency of the heart, Roche et al. developed a soft VAD based on creative use of pneumatic actuators [24], which has been leveraged in seemingly non-relevant applications such as rehabilitation [25,26] and gastric surgery [27,28]. We believe a broad view of emerging actuating solutions could be more efficient for researchers to narrow the choice down to a best fit.

This review recapitulates recent advances of actuators in four major application categories, including cardiovascular, gastrointestinal (GI), drug delivery, and micromotors. Given the wide scope of actuators, we highlight devices that can be permanently or temporarily implanted. The current work aims to convey information for a rapid selection of actuators from diverse subcategories. Accordingly, for each type, we selectively showcase the latest and representative researches, and direct readers to reviews on specific actuating types such as magnetic [29,30], electroactive [19,31], photonic [32], and thermal [33] for technical details. We note that our choice of a “wider scope, less resolution” framework in limited space of a review best fits readers who might be struggling with device design due to lacking the background knowledge about appropriate actuators. Therefore, the current work should be taken as a broad-view map over advances on diverse actuators for implantable biomedical devices.

## 2. Actuators for Various In Vivo Biomedical Applications

The four mega categories of in-body actuator applications are cardiovascular devices, endoscope and surgery-assistant devices, drug-delivery devices, and micro-swimmers (Figure 1). Additionally, there are emerging materials and strategies that are likely to foster future implementations even the current prototypes remain preliminary. Figure 1 provides a landscape of promising actuators for in-body uses, where drug delivery and cardiac/cardiovascular occur as two major subfields. Actuators used in GI capsule robots/patches can be as simple as springs or balloons, yet there are challenges on auto-triggering designs. To address this, various methods have been explored, such as pH-sensitive or glucose-responsive coatings [34]. Hydraulic actuators are preferred for endoscopes as fluid lines are typically embedded in existing endoscopes. Conventional brush or brushless motors remain the primary locomotive solution for the best control of forward/reverse camera motions despite emerging options such as magnetic and micromotors. Operating a magnetic drug-delivery device often requires an external magnet to drive the vehicle toward the target area, typically assisted by a monitor to track and guide its in-body locomotion [35,36]. This scenario implies magnetic and hydraulic actuator-based devices are ideal for surgical-assistant tools or in-clinic drug administrations monitored by healthcare professionals [37,38,39,40].

### 2.1. Actuators for Cardiac and Cardiovascular Devices

The heart is the major in-body actuator supporting oxygen supply and metabolites. The Worldwide prevalence of heart failure (HF) is 63.34 million cases, accounting for a total global financial cost estimated at $346.17 billion per year [74]. HF patients suffer insufficient blood flow due to reduced ventricular function, which is caused by either incomplete muscle squeeze or decreased ventricle volumetric capacity [75]. Repairing cardiac dysfunction by leveraging artificial soft actuators is conceptually straightforward, and the related engineering efforts are particularly valuable for patients before heart translation. One unfortunate fact is that many HF patients on the waiting list die from end-stage heart dysfunction because donor organ availability is far below the need [76]. As one alternative rescue solution, ventricular assistance devices (VADs) have been developed as a bridge until transplantation is conducted or as a permanent solution to restore cardiac functions in case of persistent organ shortage [77].

From the inside to the outside of the heart, external VADs aim at direct cardiac compression for augmenting blood flow [78]. As an artificial muscle layer, these devices assist ventricular compression without contacting blood, which forgoes the need for blood-thinning agents and their associated risks. Moreover, external VADs can be placed away from coronary vessels and other risky sites. One McKibben-based pneumatic VAD cuff was developed by the Walsh Group and tested in vivo, shown as Figure 2a [41]. Their observations confirmed that reliable device–tissue interfacing and the appropriate tuning of the contraction rate are critical to an optimal cardiac output. Upon being coupled with the heart, a systolic period of 40% achieves the highest aortic flow rate of 2 L min^−1^. The results also suggest an improved refilling function of the heart during diastole. Its pneumatic pumping and control can follow the convenient method of the existing FDA-approved pneumatic SynCardiaTM system [79]. Shortly after, the team attempted to address the device-tissue coupling problem by adding an inflatable anchor clamping the interventricular septum, shown in Figure 2b [42]. The bracing assembly consisted of a bracing bar that passes through a ventricle wall and a semilunar bracing frame around the ventricle with the integration of pneumatic actuators. In this setting, force sensors are included to provide real-time monitoring of compression forces, a critical leap toward soft VAD robots. Their next advance is a soft robotic sleeve that combines compressions and twists (Figure 2c) [24]. This combined actuation mechanism mimics the operation of a natural heart where layers of multiple linear contractile filaments are orientated along helical and circumferential patterns [80,81]. The biomimetic sleeve can be equipped with a control system designed to synchronize its actuation with the native cardiac cycle. This allows the fine-tuning of the output force and the timing of disease-specific needs. Pneumatic actuators in the devices above leverage a wall-compressed air supply for actuation. Ongoing efforts are focused on wearable pumping and control so that the on-heart actuator can work in a similar way to a more mature device [78]. Size, weight, biocompatibility, and durability are among the major remaining challenges that are currently preventing its practical use [82,83]. Nevertheless, from a more positive perspective, we should underscore that the power and control system required here are no more complex compared to the existing VADs, and that its improved mechanical match with tissues at a more reasonable cost proves unreplaceable by mechanical actuators.

The dimension for an actuator drops below the millimeter level from the heart to vessels. Medical catheters are widely used for endovascular surgeries treating illnesses such as cerebral aneurysms [14,84]. Actuators for catheters are born with a genetic advantage because the control/power stimuli (e.g., hydraulic or pneumatic) can be delivered along the catheter itself. One major challenge is that the tips of conventional catheters generally lack dexterity and are typically operated by skilled healthcare professionals under X-ray fluoroscopy [43]. To address this issue, manually operated wires [44] can be embedded in the catheter wall so that the bending can be controlled by an external electromechanical device (Figure 2d). Replacing wires with water pressure, a hydraulically steerable catheter (Figure 2e) can be embedded with four 50 μm wide fluid sub-channels that are uniformly arranged in the wall of a 0.9 mm-diameter catheter [45]. Infilling one channel triggers an expansion and bends the catheter toward the opposite side; in a similar regime, infilling two adjacent channels results in a combined bending at 45°. As such, the usability of such a configuration has been verified through tortuous cerebral vasculature and by deploying coils, and the catheter successfully accessed the ascending pharyngeal artery and parotid artery in ex vivo studies. The electrothermal input is another type of stimulus used for steering control. Selvaraj et al. recently developed a proof-of-concept catheter based on thermal-responsive hydrogel [43]. Here, repetitive bending is controlled by heating an integrated planar copper coil at the 5 mm-wide free end (Figure 2f). Note that at room temperature the catheter tip is fully curved. Specifically, bending is triggered at a critical temperature, around 28–32 °C; at a power of 3.5–4 W; and reaches a bending angle of 170° at 50 °C. To prevent the influence of body temperature, the tip needs to be encapsulated in thermal insulation. Bilateral bending remains a challenge for this design as only one-side is attached to the heating coil. Furthermore, the down-scaling of this device is feasible when the critical temperature can be tuned slightly below body temperature so the tip can hold a desired deformation to save power.

Stenting is a common solution to severe atherosclerosis caused by progressive plaque buildup on the arterial wall. Restenosis is one leading cause of stent dysfunctionality and the requirement of surgical intervention. Integrated pressure sensors and remote stent heating have been reported to detect and prevent restenosis in vivo [85,86], while controllable re-expansion proves an effective route to eliminate vessel re-narrowing risks. Shape memory alloy (SMA), as a thermoactivated material, induces re-expansion when the stent’s resonant frequency matches an external RF trigger signal [46,87]. In the example shown in Figure 2g, a 2 mm-diameter nitinol SMA stent expanded to 3.2 mm in diameter under 11.7 W RF of power at 315 MHz within 220 s, or re-expanded to 4.2 mm under 29.5 W in 100 s. The large, controllable expansion suggests its usability as a durable implant without re-intervention or re-stenting procedures. Moreover, thermal actuation based on wireless heating proves to be effective to mitigate hyperthermia as well (Figure 2h), where temperature and force sensors offer closed-loop manipulation of smart stents [88,89].

### 2.2. Actuators for Endoscope and Surgery Assistance

Endoscopes are wired or wireless GI devices that provide combined capabilities of image capturing, biopsy sampling, and surgical interventions [50]. Wired endoscopes can be inserted from natural orifices such as the rectum or mouth. The cable is typically in the range of 8–12 mm in diameter and consists of multiple sub-channels of optical or electric paths as well as fluid lines for camera flushing and GI tract inflation. An external controller with knobs is maneuvered by skilled professionals for steering, flushing, and imaging (Figure 3a). Conventional miniatured actuators based on micro-electromechanical systems (MEMS) are used for tissue sampling and surgical purpose but this often requires operation at a high voltage or temperature [90]. Russo et al. addresses this challenge by developing a low-cost fluid-driven robotic arm that enables safe interaction with surrounding tissues, shown in Figure 3b [48]. The team proposed a “soft pop-up regime” to offer sufficient force and gentle interaction with GI tissues. Their design leverages existing fluid lines to inflate/deflate a hemispherical microballoon joint for soft arm operation. One engineering advantage is that this soft fluidic microactuator (SFMA) can be fabricated in large batches [91]. Endoscope manipulation can also be assisted by an pneumatic actuator, shown in Figure 3c, which offers better mechanical matches with tissues thereby reducing the risk of accidental damage [92].

To avoid wiring issues, wireless capsule endoscopes (WCE) have been developed toward convenience operations and patient comfort since 2000 [93]. Unlike tethered probes, catheters, and endoscopes that struggle to reach the small intestine, WCE run through the entire GI tract with minimum human intervention and discomfort for minimally invasive diagnosis of unknown abdominal pain, GI hemorrhages, small bowel tumors, and Crohn’s disease [94]. The revolutionary features above make WCE a gold standard as small-intestine endoscopes where biopsies and active locomotion are not desired. A WCE passes through the GI tract within 24 h, captures images at a frequency of 4–6 frames per second, and transmits data to a wearable recorder via electric-field propagation or radio-frequency connection [95,96]. Despite features above, one major limitation of conventional WCEs is the absence of controllable locomotion. This has spawned explorations in advanced wireless actuation, which in turn fosters the evolution of in-gut robots. As to the choice of actuator, brushed/brushless DC motors have been leveraged for actuating legs and propellers in a WCE for forward and backward motion [47,49,97]. Adding one on-board magnetic actuator and then navigating the capsule with an external magnet can accelerate the capsule’s motion, which renders a favorable combination of rapid locomotion and precise actuation of robotic arms [98] for facile tissue manipulation. Sensors including pH, pressure, temperature, and gas-molecule detectors can be further integrated in such a capsule [99,100], updating it into a multifunctional robotic platform.

Actuators can provide surgical assistance under wired or wireless control. The use of an electromagnetic field is a widely reported approach to operate magnetic actuators in the GI tract. The orientation and magnitude of the external magnetic field can be controlled, and accordingly, in-body microrobots can be manipulated in closed spaces by loading X-ray contrast agents (e.g., Lipiodol) in micro-actuators [101]. Lipiodol-loaded, visualized microrobots allow easy targeting and retrieval owing to the hydrophobic properties of the Lipiodol agent, and one example is shown in Figure 3e [102]. Moreover, being observable enables flexible actuator manipulations including rotation, lifting, and flipping. The fine motion control of magnetic actuators makes it possible to precisely locate and even tune the force exerted on the inner wall of the intestine [40], uterus [103], and stomach [104,105]. Hwasaki et al. reported a soft patch remover driven by a magnetic actuator, shown in Figure 3d [106]. The remover is navigated to the target (stomach patch) and compressed firmly against the target to create negative pressure (pseudo-vacuum). Next, one side of the patch is lifted by the suction cup to peel it off. This combination of magnetic actuators and X-ray or ultrasound-imaging techniques proved an effective approach to press microneedles into the wall of the intestine and uterus for drug delivery, embryo transfer, and tumor surgeries, the details for which can be found in recent reviews on magnetically controlled soft robots [107,108,109].

### 2.3. Actuators for Drug Delivery

An oral drug administration is often preferred over injection as it is free of needle-associated pain and safety concerns. For biomacromolecules such as insulin and adalimumab, however, subcutaneous self-injection remains the gold standard because such biologics have difficulty penetrating the barrier of the GI tract [111]. Exploring oral solutions for such drugs is of tremendous value. For example, the immunosuppressive drug adalimumab (e.g., Humira^®^) reached a global sale of $20.7 billion in 2021 [112]. The huge market and user preferences have inspired flaring interest in academia and industry. Generally, altering the injections of oral administration requires novel drug-delivery vehicles being able to land on a preferred site and delivering biologics across the GI barrier. This can be realized by a 5–8 mm long canular to penetrate the stomach’s wall (about 5 mm in thickness), or by much shorter microneedles (length < 1 mm) in the small intestine [113]. Inspired by the self-orienting capability of the leopard tortoise, Abramson et al. [52] recently designed a self-orienting drug delivery capsule that can resist external forces arising from fluid flow or peristatic motion once it is attached on the stomach wall (Figure 4a). The drug-containing millipost is inserted by a hydration-respondent spring actuator, where its trigger is encapsulated by dissolvable sucrose. This setting reserves vents in the capsule to trigger the dissolution of the sucrose/isomalt protective coating in GI fluid thus releasing a small spring. Similar designs may find more in vivo applications as the sucrose dissolution time can be fined tuned with a precision of 11.4 s. The microneedle-based solution from Rani Therapeutics, San Jose, CA, leverages a mini-balloon that can be inflated by chemical reactions, as shown in Figure 4b [114]. This pill has an enteric coating that protects itself in stomach acid. When pH levels rise as it arrives at the small intestine, the coating dissolves and triggers a chemical reaction to inflate the balloon, which eventually pushes dissolvable microneedles to release the drugs in the GI barrier’s layers. A similar pH-sensitive actuation mechanism has been implemented in a self-unfolding microneedle-based drug-delivery patch, which can load insulin and other biologics [53]. Figure 4c demonstrates a gastric-resident electronic system with two arms to extend its residence in the stomach [54]. The arms are self-expanded upon exposure to stomach acid and will detach from the drug-delivery module after 36 days use—the disintegration of which allows safe passage from the gastric space to intestine. This system can load commercially available modules for drug-delivery, sensing, and sampling tasks. More smart pill devices for GI diagnostics and therapy are critically compared in other reviews [115]. Often, the major challenge is the design of an auto-triggering mechanism to activate the actuator at the target GI site.

Drug-delivery pumps designed for the in-body environment necessitate biocompatible and power-efficient actuators. One most disruptive solution is the use of natural muscles, e.g., worm or insect muscles. Earthworm muscle, shown in Figure 4d, has been evaluated for controllable drug delivery by Tanaka’s group [116]. The natural combined mechanism of the longitudinal and circular actuations renders a more favorable laminate geometry compared to skeletal muscles [121,122,123]. Their pump achieved a flow rate of 5.0 μL s^−1^, which is about 3–4 orders higher than a similar form based on a cardiomyocyte pump [124,125]. Artificial muscles (AMs) based on electroactive polymers (EAPs) demonstrate large deformation through ions/cations’ exchange with surrounding electrolyte fluids (e.g., saline) at low voltage and small power consumption [126,127]. The precise tailoring and engineering of AMs are challenging to date due to their chemical stability against most chemicals including lithographic acids [128,129]. Very recently, laser cutting, as an automatic route, has been reported to realize various actuations including lifting, pulling, rotation, and squeezing [130,131]. A polymer squeezer fabricated via a laser approach proved capable of actuating a battery-free, implantable insulin pump at a small power of 2 mW, which can be delivered wirelessly by a thumbnail-sized antenna, shown in Figure 4e at the bottom [56,117]. Such biocompatible, power-efficient, and soft polymers are becoming engineerable actuator materials, though the 3D structuring of EAPs remains a challenge. Another type of EAP, named ionic polymer-metal composites (IPMC), is particularly promising for in-air use [55]. A solid-state electrolyte gel in an IPMC actuator is sandwiched into an opposite electrode (working and counter) during electro-actuation [132]. This setting allows for the voltage-controlled bending of the multi-layered film. Forgoing an electrolyte solution implies that such devices, after proper miniaturization, may find wider applications, for example, a diaphragm valve (Figure 4e, top) [118].

A piezoelectric (PZT) film is another major type of electroactive material for implantable drug delivery. Compared to EAPs, a PZT possesses a higher frequency (beyond 1 kHz) and longer lifetime (over million cycles). Another feature is that a PZT can perform actuation and sensing at the same time, thereby offering an attractive feasibility for closed-loop control [133,134]. The limitations of PZT are a smaller strain and relatively higher driving voltage (typically above 100 V without material modification). The other challenge is biosafety concerns due to lead leakage from PZT, and this has been addressed by multiple coating methods in the past decade [135,136,137,138].

A catheter itself can pump out fluid when one photodeformable layer is embedded. Xu et al. recently reported a microtube featuring a liquid manipulation ability driven by a photo-responsive layer of azo linear liquid crystal polymer (LLCP) [120]. Under 470 nm light with 80 mW cm^−2^ intensity, a liquid slug can be manipulated in both straight and curved stages. Photodeformable azo LCPS provides fast and tunable deformation so their implementation in catheters may simplify microfluidics systems significantly through a rational combination of contraction/expansion, bending, twisting, and rolling actuation [32,139].

### 2.4. Actuators in Bio-Hybrid Robots

In-body micro-actuators under 100 μm require a revolutionary *hybrid* design. Microorganisms including cells and bacteria can be implemented as the natural propellers on this scale to drive a functional load towards a target by various means including light, magnetism, electricity, or a chemical gradient [62,140,141]. High-motility microorganisms, such as *Escherichia coli* (*E. coli*), *Salmonella typhimurium* (*S. typhimurium*), *Serratia marcescens* (*S. marcescens*), etc., are favored for their high-speed, typically over hundreds of their body lengths per second, allowing a free motion in capillaries and interstitial area [142]. Figure 5a demonstrates bacterially propelled drug-delivery robots based on *S. typhimurium* developed by Park et al. [143] In this setting, attenuated strains of flagellar bacteria have unique advantages in delivering cancer therapies because they can specifically target and proliferate in tumors [144,145]. Their hybrid drug-delivery vehicles leave the vasculature actively and penetrate into deep tumor tissue. Other benefits include their capacity for sensing, moving, accumulating, and replicating in solid tumors [146,147]. Felfoul and his team developed a similar hybrid swimmer based on the Magnetococcus marinus strain, MC-1, to transport drug-loaded nanoliposomes into hypoxic regions of the tumor (Figure 5b) [148,149]. Guided by a magnetic field, MC-1 cells’ tendency to swim to low oxygen areas are as a result of its two-stage aerotactic sensing system [149]. The results revealed up to 55% of MC-1 cells can penetrate hypoxic regions and into colorectal xenografts. As another example, sperm cells, as a best known micro-swimmers, can also be re-configured for drug-delivery purposes (Figure 5c) [36]. Sperm cells do not proliferate, form colonies, or express pathogenic proteins, which makes them a promising vehicle to deliver anticancer drugs in the female reproductive tract for the treatment of cervical cancer and gynecologic disease [35,64,149,150]. Encapsulating hydrophilic drugs is not a challenge as sperm cells have high DNA-binding affinity [151] and thus can store drugs in their crystalline nucleus [150]. With this setting, the membrane itself is a protective barrier against immune reactions and enzyme-induced degradation. A tetrapod trap is designed to capture the sperm’s head. Upon hitting the targeted cell cluster, the tetrapod’s four arms will bend, thereby releasing the drug-loaded sperm cell.

In addition to cells and bacteria, micro-swimmers can be propelled by magnetic force or natural muscles. A magnetic clot-remover, shown in Figure 5d, can remove blood clots at a maximum rate of 20.13 mm^3^ per minute [152]. The swimmer is 2.5 mm in diameter, 6 mm in length, and has a cutting tip coated in diamond powder. Its permanent magnet allows the swimmer to be propelled by an external magnetic system using three coil pairs arranged orthogonally. Similar swimmers can reach a velocity of 100 mm s^−1^, which enables counter-flow navigation in arteries outside the heart and aorta [153]. The swimmer’s size nicely matches the conventional catheter insertion, but an in-body use requires considering challenges including imaging, system latency, and extra resistance arising from blood flow. Muscles, as highly contractile actuators, are electrically responsive to a small triggering potential, which favors electronics integration for onboard powering and control [154]. Skeletal muscles have been massively explored for biosystems as well [155,156]. Culturing skeletal muscles requires substrates that are biocompatible, conductive, and of limited inherent rigidness. Figure 5e demonstrates a muscle-based device with a substrate of a conducting polymer (CP) film/coating [72,73]. It should be pointed out that CPs offer an intriguing combination of desirable features for cell culture and have been used as a major scaffold for muscles and neurons [157,158,159]. CP films, for example, polypyrrole, can maintain a high electrical conductivity over 100 S·cm even at a small thickness below 300 nm, which allows easy control of their actuation [73]. Worm-like actuations are observed when flexible hinges are preserved along the electroactive stripe. Unlike CP-based actuators relying on external stimuli, actuators can be self-controlled by on-board neural commands. Aydin et al. [68] recently disclosed a muscle-powered swimmer machine piloted by on-board neuromuscular units, shown as Figure 5g. The hybrid power chain involves skeletal muscles that are cultured in situ with optogenetic stem cell-derived neural clusters containing motor neurons. Encapsulated by polydimethylsiloxane (PDMS) and driven by light, the swimmer achieved a speed of 0.7 μm s^−1^. Although this light-based micromachine may not be used in the body at present, this work highlights the concept of potential responsive microsystems. Insect muscles are among the toughest natural actuators on earth as they can tolerate a wider range of external and internal conditions than birds, mammals, and vertebrate ectotherms [122,160]. One pair of micro-tweezers based on insect muscles was developed by Akiyama et al. [71]. The device is fully packaged by a biocompatible and mechanically robust coating so as to operate outside of a culture medium. Insect muscle is also a good candidate to propel swimming micro-robots, as insect muscle is more robust in diverse environments compared to mammalian muscle cells. Figure 5f shows one of the micro-robots driven by insect muscles [121]. The autonomous swimming robot retain functionality at room temperature without pH or temperature maintenance. The swimming speed of 11.7 μm s^−1^ is slow, but it can work, on average, for two months or even longer at 0.15 Hz [161]. Muscle-based actuators and robots remain preliminary compared to other types of actuators, and their dimensions must be reduced to fit catheters and other surgical appliances. Biocompatible encapsulation is another concern, as contracting cells require nutrition/oxygen supply from a culture medium. Despite the challenges above, being autonomous is favorable for micro-actuators working in the body. One potential use of such a muscle-based actuator is as a steerable catheter as they can bend or vibrate at a constant rate without external control.

## 3. Discussion

### 3.1. Biocompatibility

The biocompatibility of actuators refers to the ability of their short- or long-term operation in the body with an appropriate host response [162]. An actuator can be inherently biocompatible or requires external coating to meet biocompatibility requirements. Inherently biocompatible actuators are developed based on bio-safe materials such as medical stainless steel, SMAs (e.g., NiTi), and shape memory polymers (SMPs). Molecular modifications on the material level have proven to be effective for improving biocompatibility. One such an example is the doping of macromolecular counterions in EAPs. Small counterions (e.g., BF_4_^−^) doped in EAPs may diffuse out of the polymer matrix into the surrounding tissue during operation, which triggers cytotoxic, oxidative, or genotoxic effects in the specific environment. [163] Macromolecular counterions such as TFSI [164] and polyol-borate [165], on the other hand, stay entangled with the host matrix, thus significantly improving the biocompatibility of the host polymer. Actuators can still be implemented in vivo even when the material is toxic. This is a case where biocompatible coatings should be considered to encapsulate either the actuator or the entire device. There is a rapidly expanding library of bio-safe coatings that can be deposited physically [166], chemically [167], or electrochemically [168]. Some existing reviews have investigated metals, hydrogel, and polymers as biocompatible interface [135,169,170,171,172,173,174]. It should be noted that no actuator is biocompatible in all environments; SMAs that are safe as a stent material may still induce biofouling effects when serving as an implantable neuron interface [175], so the biocompatibility of actuator/coating materials should be analyzed in specific settings.

### 3.2. Powering

Fully automatic in-body robots based on integrated lithium batteries, biofuel cells [176], nuclear power [177], and electrostatic [178] batteries have been a longstanding technical pursuit, yet self-powering is neither necessary nor feasible for most applications to date. By contrast, existing devices depend on external power that can be delivered in either a wired or wireless manner. Wired endoscopes, catheters, and short-term VADs are convenient for minimally invasive surgeries as the cord itself can be used to conduct electric, optical, fluidic, or air power.

The most widely reported wireless-powering approach is the use of an electromagnetic field, whose magnitude and direction can be readily programmed for actuator operation [179,180,181]. Wireless electrical power, which is particularly promising for battery-free implantables, can be achieved through an implanted energy harvester that converts RF energy collected to mW-level DC output [182,183,184]. Implantable devices can also be powered by other sources including photovoltaic [183], ultrasonic [185,186], kinetic, and thermoelectric energies from the body [187,188]. Note that power should be considered together with the selection of the actuator at the very beginning of device design because different powering strategies may massively alter the form factor and patient adherence. This process involves the careful evaluation of human factors as well as the technical feasibility. Magnetic power, for example, could interfere with adjacent devices [189,190], so the intensity should be controlled to mitigate such issues. Novel powering strategies based on advanced batteries or wireless coupling have triggered intense interest and more details can be found in other reviews [177,178,191,192].

### 3.3. Recommendation

Choosing an actuator can be simplified with more knowledge on the advantages and limitations of each type of actuator, a summary of which is showcased in Table 1. Note that actuators, even within the same type, may demonstrate remarkably different performances (e.g., strain/stress output) when their structure or size changes [57,193], so readers should use the data in this table to estimate the typical ranges but not constant values. Breaking down into the specific type, pneumatic actuators can provide strain over 300% but require wired control (air pressure) and are suitable for surgical-assistance tools rather long-term implantables [14,77,194,195]. EAPs, particularly those embedding macromolecular counterions as dopants [196,197], stand out with a high strain output, small power consumption, and excellent biocompatibility, and, therefore, can be a promising candidate for steerable catheters, soft valves, and battery-less pumps. Care should be taken as EAPs are generally unsuitable for applications requiring high force/stress or fast response [198,199] since their deformation is induced by slow voltage-driven ion exchange. SMAs and miniaturized spring/balloon actuators offer a large force output, but their strain/deformation is typically programmed before implantation and cannot be adjusted by external controls thereafter. To date, most bio-hybrid actuators are ex vivo prototypes whereas leveraging a bio-hybrid device an in vivo setting remains difficult as the device itself needs closed-loop control and self-powering [35,59,64,65]. At last, we highlight biohybrid actuators as they are constructed at the micro/nanoscale based on bacteria, myocytes, and cultured/harvested insect muscles free of electronics and batteries, which implies that some biohybrid actuators may possess unique advantages as in-body swallowable vehicles that can safely dissolve in the GI tract.

## 4. Conclusions

This study offers a broad overview of the established and potential connections between novel actuators and their in-body applications in cardiovascular devices, endoscopes, drug delivery capsules, steerable catheters, and micro-swimmers. Generally, in-body devices prioritize usability and safety (biocompatibility, size, and cordless operation) over force, deformation, or frequency. Moreover, the specific physiological environment should be considered for biocompatibility assessment because, for instance, anchoring on the thick stomach wall may require a long stroke needle, which could be dangerous when applied in the small intestine. We thus suggest three principles for actuator’s design: (1) the actuator must be safe for the targeted use, with no or only minimal biofouling or cytotoxicity; (2) the actuator should provide sufficient strain, force, and speed to fulfill a given need throughout the device’s lifetime; and (3) the powering and control of the actuator should be realized in a manner that will not significantly compromise patient adherence. The increasing technical readiness of implantable actuators forecasts a future wherein implantable robots will be developed on the macro- and micro-scales.

## Figures and Tables

**Figure 1 micromachines-13-01756-f001:**
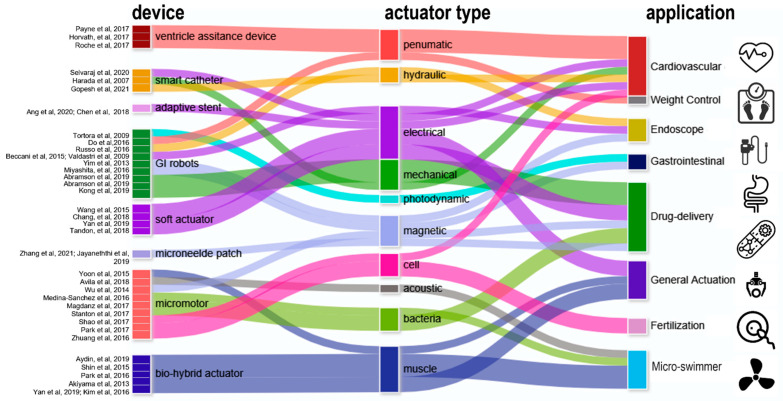
Actuators in various biomedical devices for major in-body applications [24,37,38,39,40,41,42,43,44,45,46,47,48,49,50,51,52,53,54,55,56,57,58,59,60,61,62,63,64,65,66,67,68,69,70,71,72,73].

**Figure 2 micromachines-13-01756-f002:**
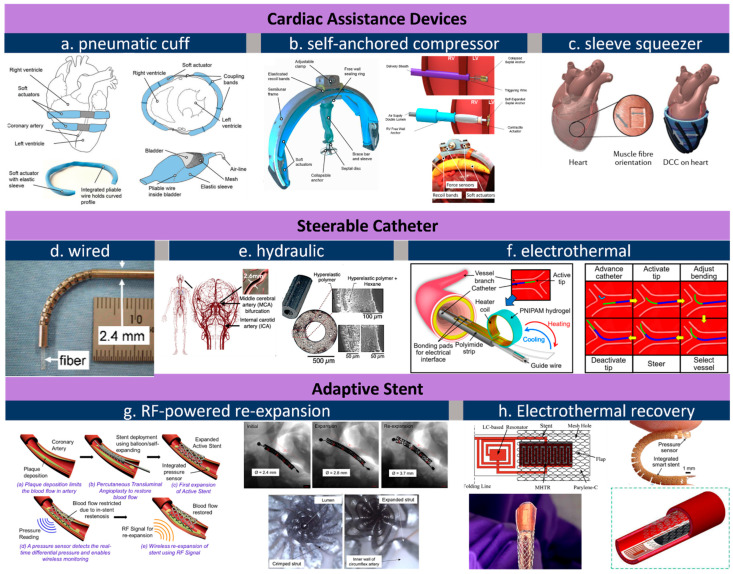
Actuators for cardiovascular applications. (**a**), pneumatic cuff that is wired to heart’s contour [41]; (**b**), a self-anchored assistance-squeezer [42]; (**c**), a pneumatic sleeve for twist-n-squeeze actuation [24]; (**d**), steerable catheter through wired control [44]; (**e**), a fluid-steered catheter [45]; (**f**), electrothermally controlled steering of catheter tip [43]; (**g**), RF-triggered re-expansion against restenosis concept (left) and its in vivo image before and after RF expansion (right) [87]; (**h**), electrothermal treatment of restenosis [89]. All images are reproduced or adapted with permission.

**Figure 3 micromachines-13-01756-f003:**
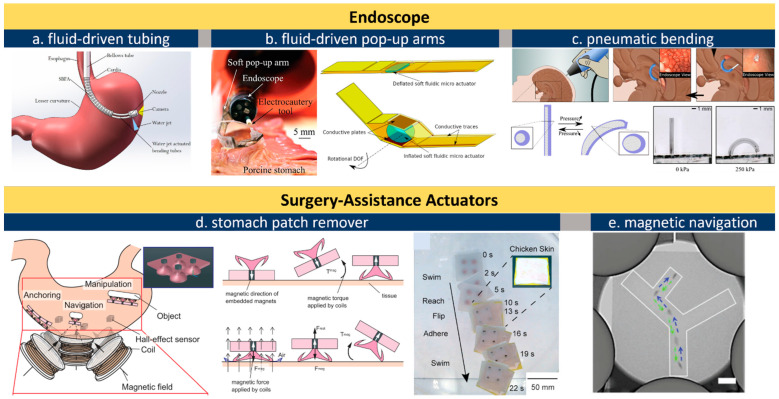
Actuators for endoscopes. (**a**), schematic illustration of the fluid line in a wired endoscope kit [110]; (**b**), an endoscope equipped with soft pop-up arms [44,48]; (**c**), pneumatically inflated actuation of intracranial endoscope [92]; (**d**), a magnetic soft vacuum sucker to remove surgery patch in stomach [106]; (**e**), X-ray navigation of magnetic actuators under skin [102]. All images are reproduced or adapted with permission.

**Figure 4 micromachines-13-01756-f004:**
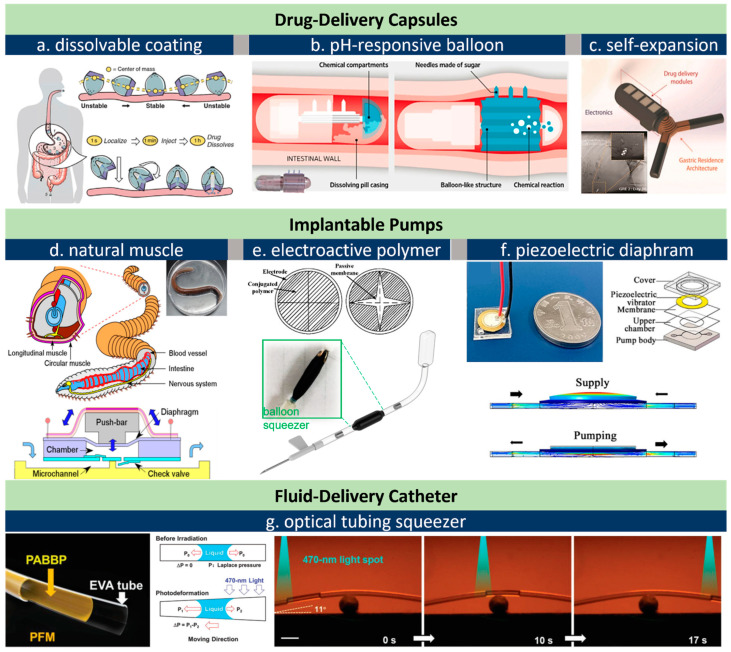
Actuators for drug-delivery uses. (**a**), Spring actuator triggered by dissolvable coating in a tortoise-mimic capsule [52]; (**b**), pH-responsive inflatable balloon actuator for controlled microneedle insertion in RaniPill^TM^ [114]; (**c**), self-expanded arms in a stomach-residing capsule [54]; (**d**), a drug-delivery pump based on worm muscle [116]; (**e**), EAP-based implantable insulin pump (bottom) [117], and an EAP-based diaphragm valve (up) [118]; (**f**), a piezoelectrically driven implantable pump [119]; (**g**), photoresponsive catheter for fluid/drug delivery, scale bar is 2 mm [120]. All images are reproduced or adapted with permission.

**Figure 5 micromachines-13-01756-f005:**
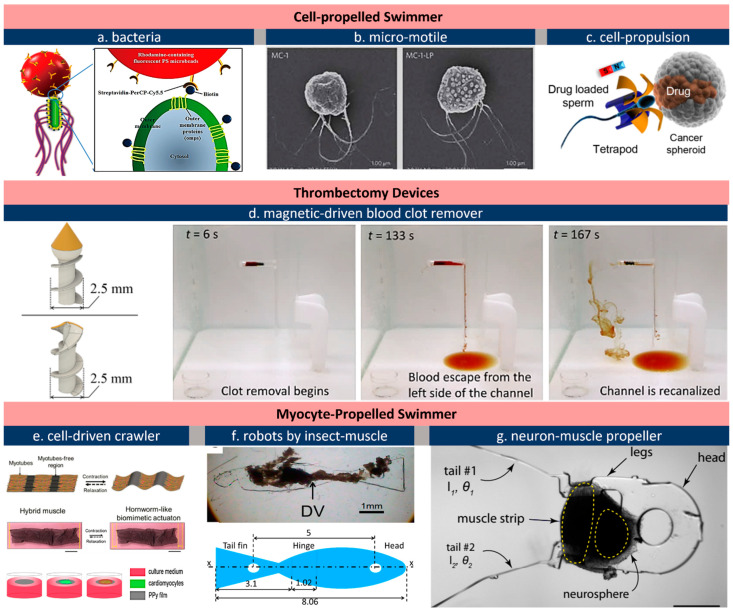
Bio-hybrid actuators for in-body locomotion and general actuation purposes. (**a**), Bacterially propelled micro-swimmer [143]; (**b**), magnetically guided micro-motile for drug delivery to tumor cell [148]; (**c**), sperm-propelled drug delivery vehicle [36]; (**d**), a clot remover driven by external magnets [152]; (**e**), free-swimmer based on skeletal muscles cultured on ultrathin (up) and electrically conducting substrate (bottom) [72,73]; (**f**), a micro biomimetic fish driven by insect muscle [121]; (**g**), cardiomyocyte-propelled swimmer controlled by on-board co-cultured neuron unit [68]. All images are reproduced or adapted with permission.

**Table 1 micromachines-13-01756-t001:** Specifications of Different Types of In-body Actuators.

Specifications	Pneumatic	Fluidic	Electric	Magnetic	SMA	Biohybrid
Electrothermal	EAPs	Piezoelectric
Typical Size	Macro 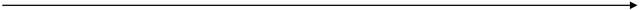 Micro
Strain	high, can be >300%	5–20%	30–40% with modification (CNT) ^N6^	>40%	<10%	design dependent	1–10%	<20%
Frequency	0.5–60 Hz	<0.1 Hz	<0.1 Hz	0.05–1 Hz	Up to >100 kHz	up to >100 kHz	0.1–35 Hz	1–3 Hz (cardiomyocyte)
Young’s Modulus ^N4^	soft (<100 kPa)	medium rigid (<1 MPa)	soft	soft	rigid (>1 MPa)	soft	rigid	soft
Powering	air pressure	fluid pressure	3–80 mW, 4.5–40 V	2–20 mW, 0.7–2 V	5–1000 V	electromagnetic, >100 mW	electric, >100 mW	<1 mW ^N1^
Control ^N5^	wired	wired	Wired/wireless	wired/wireless	wired	wireless	wired/wireless	wireless
Biocompatibility	medium	medium	high	high	low ^N3^	medium	medium	medium
Efficiency	<20%	40–55%	TBD	>30%	<30%	80–90%	>4%	TBD ^N2^
Lifetime (cycles)	>10^6^	NA	10^7^	10^3^–10^6^	>10^9^	>10^6^	<10^4^	NA
Advantages	Large deformation	Large strain and force, compatible with endoscope	Large strain and force	Biocompatible, low power consumption	Precise strain control (0.1 µm resolution), large force	High speed, large force, programmable strain	Large deformation and force, biocompatible	Fewer requirement for batteries and electronics
Limitations	Large size, requires pressure pumping	Requires fluid pumping, not as lightweight as others	Slow actuation, difficult strain control/hold, thermal interference with surround tissues	Slow yet inevitable loss of capacity for ions-exchange due to electrochemical instability	Can be not safe for in vivo operation without protective coating, relatively high voltage	Requires external magnetic field for navigation	Requires external stimuli (thermal, optical, etc.)	Technical readiness for in-vivo applications remains low
Best for	surgical tools	endoscope arms	Hand protheses, rehabilitation-assistance	valves, pumps	valves, pumps, energy harvester	locomotor/pumps	durable stent	TBD ^N2^
Reference	Payne 2017 [41]Horvath 2017 [42]Roche 2017 [24]Do 2016 [37]	Gopesh 2021 [45]Russo 2016 [48]	Tian 2021 [200]Yin 2020 [201]Potekhina 2019 [33]	Wang 2015 [55]Chang, 2018 [61]Yan 2019 [131]Tandon 2018 [57]	Shan 2022 [119]Gao 2020 [136]Nafea 2018 [202]Choris 2019 [137]	Yim 2013 [38]Miyashita, 2016 [51]	Song 2016 [203]Liu 2021 [169]Shull 2018 [175]Ang 2020 [46]	Aydin, 2019 [68]Shin 2015 [69]Park 2016 [70]Kim 2016 [72]

Note: N1. Power supply for bio-hybrid actuators is typically very small below 1 mW, specific number needs further studies. N2. Efficiency and applications of bio-hybrid actuators remain to be determined (TBD) given its presently limited technical readiness of this relatively novel concept. N3. Lead-enriched piezoelectric materials require coating or surface modification to improve biocompatibility. N4. Young’s modulus for pneumatic, hydraulic, and magnetic actuators depends on the host materials. N5. The term “wired” includes both cable-delivered power and on-board battery power. N6. Carbon nanotube (CNT) is doped to enhance strain output.

## Data Availability

Not applicable.

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
