# Peer review of "Actuators for Implantable Devices: A Broad View"

_micromachines, 2022, doi:10.3390/mi13101756_

Round 1

Reviewer 1 Report

The review provides a summary of novel applications for micro systems in the body. Many tyypes of micro actuations are not mentioned and discussed such as, piezoelectric actuation (See Abadi, A. and Kosa, G., 2016. Piezoelectric beam for intrabody propulsion controlled by embedded sensing. IEEE/ASME Transactions on Mechatronics, 21(3), pp.1528-1539.) magnetic (the seminal works of Brad Nelson) micojets (J. Li, B. E.-F. de Ávila, W. Gao, L. Zhang, J. Wang, Sci. Rob. 2017, 2,

eaam6431),

Please mention other reviews made on the topic such as Kosa, G. and Hunziker, P., 2019. Small‐Scale Robots in Fluidic Media. Advanced Intelligent Systems, 1(7), p.1900035. This is comparative study that uses scaling.

The English should be improved. The quality of the figures and tables is not good enough.

Reviewer 2 Report

Unfortunately I can not change my opinion of rejection, as the manuscript, despite some additional paragraphs, is equivalent to the former version.

Author Response

NA (No revision request received)

Reviewer 3 Report

This paper presents a review of implantable actuators. The paper is interesting but requires further improvements, including the following issues:

1.    The title needs to match the contents. Not all examples presented can be considered as robots. The author may reward the title and other sections to remove the word “robot” and make these parts based on actuators only. Otherwise, non-robotic examples should be removed.

2.  Some sentences in the Abstract require rephrasing. In addition, more sentences should be included to describe the contents of the paper.

3.  Several papers have addressed this topic from different points of view. What is different in this paper? The author should compare this paper to previously published review papers on this topic. This can be done in the last paragraph of the Introduction, for instance.

4.  The author mentioned the biocompatibility in the paper several times, which is a good thing. However, since the paper focuses on implantable actuators, it is expected to include a section that is completely dedicated to biocompatibility. This section should discuss several issues related to biocompatibility, such as the selection of materials, encapsulation, life expectancy of the device, … etc. The authors may refer to the following papers, which discuss the biocompatibility issue:

·       https://doi.org/10.1002/adma.201802084

·       https://doi.org/10.1002/asia.201900292

·       https://doi.org/10.1088/1361-6439/ab8832

·       https://doi.org/10.1016/j.mattod.2020.12.020

5.    One of the essential topics that need to be discussed in this paper is the powering methods. The paper is expected to have a dedicated section for this matter since it is one of the main challenges when implanting actuators. Powering approaches can be generally categorized as on-board sources, such as biofuel cells, lithium batteries, nuclear batteries, electrostatic, …etc., and wireless sources, such as optical charging, ultrasonic, inductive coupling, … etc. The author may follow other categorization methods. There are many papers that discuss this topic, including the following:

·       https://doi.org/10.3390/s151128889

·       https://doi.org/10.1016/j.bios.2020.112781

6. Based on the previous two comments, when it comes to the life expectancy of the device and its powering method, wireless powering methods, such as inductive coupling, are considered among the future directions in this area. This topic can also be discussed in the powering methods section. The authors can refer to the following papers, including papers by John A. Rogers, who has several interesting papers on this topic:

·       https://doi.org/10.1109/ACCESS.2015.2406292

·       https://doi.org/10.1016/j.sna.2015.06.017

·       https://doi.org/10.1109/JMEMS.2017.2692251

·       https://doi.org/10.1016/j.sna.2018.06.020

·       https://doi.org/10.1063/1.5099128

·       https://doi.org/10.1038/s41586-019-1687-0

·       https://doi.org/10.1038/s41928-021-00614-9

·       https://doi.org/10.1073/pnas.2020398118

7.   There is a lack of figures that show categories, features, applications, … etc. Generally, review papers include such figures to summarize several important points and provide a better understanding of the topic. The author can illustrate a figure that includes the types of actuators discussed in this paper along with their applications, for instance. The author may refer to review papers, including the examples provided in this report, to create such figures.

8.     There is a lack of tables in the paper, which is unusual for a review paper. Table 1 is interesting, but the paper should include more. The author should consider including a table in each section to present important comparisons.

9.   The Discussion section requires improvements. The section should discuss the main challenges and future prospects in this area. The author may discuss the miniaturization possibilities, powering challenges, connectivity, security, biocompatibility enhancement, … etc.

10.  Please include Table 1 as text, not as an image. I also believe that the table should include piezoelectric actuators as they are among the most commonly used ones in implantable devices.

11.  Please select a suitable “Data Availability Statement”.

12.  References 21 and 22 are repeated. Please check all the references.

13.  Please avoid using informal English, such as the use of “doesn’t”.

Round 2

Reviewer 3 Report

The authors have addressed my comments.